



# Typhoon-associated air quality over the Guangdong–Hong Kong–Macao Greater Bay Area, China: machine learning-based prediction and assessment

Yilin Chen[1], Yuanjian Yang[1], Meng Gao[2]

[1]School of Atmospheric Physics, Nanjing University of Information Science & Technology, Nanjing, 210044, China
[2]Department of Geography, Hong Kong Baptist University, Hong Kong, China

*Correspondence to*: Yuanjian Yang (yyj1985@nuist.edu.cn)

**Abstract.** The summertime air pollution events endangering public health in the Guangdong–Hong Kong–Macao Greater Bay Area are connected with typhoons. The wind of the typhoon periphery results in poor diffusion conditions and favorable conditions for transboundary air pollution. Random Forest models are established to predict typhoon-associated air quality in the area. The correlation coefficients and the root-mean-square errors of the air quality index (AQI) and $PM_{2.5}$, $PM_{10}$, $SO_2$, $NO_2$ and $O_3$ concentrations are 0.84 (14.88), 0.86 (10.31 $\mu g/m^3$), 0.84 (17.03 $\mu g/m^3$), 0.51 (8.13 $\mu g/m^3$), 0.80 (13.64 $\mu g/m^3$) and 0.89 (22.43 $\mu g/m^3$), respectively. Additionally, the prediction models for non-typhoon days are established. According to the feature importance output of the models, the differences in the meteorological drivers of typhoon days and non-typhoon days are revealed. On typhoon days, the air quality is dominated by local source emission and accumulation as the sink of pollutants reduces significantly under stagnant weather, while by the transportation and scavenging effect of sea breeze on non-typhoon days. Therefore, our findings suggest that different air pollution control strategies for typhoon days and non-typhoon days should be proposed.

## 1 Introduction

The rapid and continuous economic and industrial development of China in recent decades has resulted in a mounting air pollution problem in the country. Major atmospheric pollutants, such as particulate matter (PM), $O_3$, $SO_2$ and $NO_2$, not only have important impacts on ecosystems, traffic safety and weather/climate, but also seriously exacerbate human health issues and increase morbidity and mortality from cardiovascular and respiratory diseases (Che et al. 2005, 2014; Zhu et al. 2021). The Guangdong–Hong Kong–Macao Greater Bay Area (GBA), located in southern China, comprises nine municipalities of Guangdong Province, including Guangzhou and Shenzhen, and two Special Administrative Regions of Hong Kong and Macao. With a high population density of over 1100 people per square kilometer, the GBA is one of China's most heavily populated and urbanized areas. As a result, the area sees a high intensity of air pollutant emissions and frequent air pollution events (Deng et al. 2008, 2011; Hou et al. 2019). As well as the intense emission of pollutants, the other main factor influencing the air quality is the weather circulation pattern (Yang et al. 2018; Zong et al. 2021). For instance, when light



breezes and a temperature inversion layer occurs in the surface layer of the GBA, the air quality deteriorates (Tong et al. 2018a; Ding et al. 2004; Huang et al. 2005; Yang et al. 2012). Relatively, the air quality is good when the wind speed in the area is high—for example, the strong southerly winds in summer and northerly winds that cross the northern mountains in winter (Chen et al. 2016; Tong et al. 2018a, 2018b).

The GBA is continually affected by typhoons in summer (Ying et al. 2014; Lu et al. 2021), and as they make landfall, the
air quality and synoptic situation in the region changes significantly (Ding et al. 2004; Huang et al. 2005; Lam et al. 2005; Feng et al. 2007; Wei et al. 2007, 2016; Yang et al. 2012; Wang et al. 2022). The causes of typhoon-associated air pollution can be concluded as follows. On the one hand, the downdraft of the typhoon periphery leads to a large-scale temperature inversion layer, meaning light air and adverse pollutant diffusion conditions prevail in the area (Feng et al. 2007; Yang et al. 2012; Deng et al. 2019). Additionally, pollutants in the upper level are transported down to the lower atmosphere, where
they accumulate under the impact of the downdraft. Consequently, the accumulation of local-source emissions is aggravated, making the air quality bad or even severe (Huang et al. 2005; Wei et al. 2016). On the other hand, the various wind patterns of the typhoon periphery (mostly northerlies during pollution events) provide favorable conditions for transboundary air pollution from both outside the GBA and cities inside the GBA (Chow et al. 2018; Lam et al. 2018; Luo et al. 2018; Deng et al. 2019; Yim et al. 2019; Yang et al. 2019).However, there are still two issues with respect to typhoon-associated air quality
in the GBA that have yet to be fully understood: 1) Which local meteorological factors play the dominant role in the change in different atmospheric pollutants during typhoon processes? 2) What are differences in the dominant local meteorological factors between typhoon and non-typhoon processes? These two issues are of great significance to the forecast of air quality and the adaptions of air pollution in the GBA.

Quantitative analysis and the prediction of pollutant concentrations have become a focus in this field of study. Existing
methods include numerical forecasting, statistical forecasting and machine learning. In terms of numerical forecasting, several models have been developed, such as CMAQ (developed by the U.S. EPA) and NAQPMS (developed by the Institute of Atmospheric Physics, Chinese Academy of Sciences) (Arnold et al. 2003; Li et al. 2011). These models have been applied by some researchers to study typhoon-associated air quality and results have revealed the impacts of meteorological conditions on the transportation and diffusion of air pollutants—for example, the downdraft, northerly winds
and high near-surface air temperatures that boost the photochemical reaction that generates $O_3$ (Wei et al. 2016). Numerical experiments also led to the discovery that the contribution of cross-regional transportation varies with the wind field, these studies reflect one of the advantages of the numerical modelling method: that they can analyze the formation mechanism of a specific pollution event (Huang et al. 2005; Lam et al. 2005). However, this approach also has its drawbacks, such as computational complexity and high data requirements. As for statistical methods, examples include clustering and multiple
regression methods based on meteorological factors and weather types (Su et al. 2009; Singh et al. 2012). Although the calculations involved in these statistical methods are simple, their predicted results exhibit uncertainties with large errors and local dependence (Ross et al. 2007; Singh et al. 2012). In contrast, machine learning methods perform very well in terms of accuracy and are already leveraged in many fields such as meteorology and the environmental sciences (Li et al. 2021,



Zheng et al. 2021, Bochenek & Ustrnul 2022, Chen et al. 2022). The forecasting of air quality is no exception. The most widely used algorithms include Random Forest (RF), support vector machines, extreme gradient boosting (XGBoost), and neural networks. The input variables include meteorological data and traffic flow data. Among the machine learning models, RF is an ensemble machine learning algorithm based on decision trees, which has certain advantages in capturing the nonlinear relationship between variables. Attempts made to employ RF in predicting air quality have produced promising results (Kamińska 2018; Bai et al. 2019; Hu et al. 2021; Ding et al. 2022; Liu et al. 2022).

It is clear from the literature, as reviewed above, that there is a definite link between typhoons and air quality in the GBA. Nevertheless, the meteorological determinants of different kinds of pollutants during a typhoon event are still unclear. There is also little research on applying machine learning in predicting air quality with typhoon location and intensity data, and the accuracy of such predictions remains unknown. Therefore, in order to improve the accuracy of air quality prediction for the GBA during typhoon processes, the present research establishes an RF prediction model of typhoon-associated air quality in the GBA with air quality data [air quality index (AQI), $PM_{2.5}$, $PM_{10}$, $SO_2$, $NO_2$, $O_3$] from 36 air quality monitoring stations in 10 cities in the region, western North Pacific typhoon tracks and intensity data from 2014–2020 derived from the China Meteorological Administration (CMA) tropical cyclone best-track dataset, and meteorological data from the fifth major global reanalysis produced by the European Centre for Medium-Range Weather Forecasts (ERA5). Also, for the non-typhoon days (NTY days) in the typhoon season (June-September), RF prediction models based on meteorological elements are established to analyze the changes in local meteorological determinants. The aim of this study is to improve the prediction and assessment of typhoon-associated air quality in the GBA, which is not only important from a scientific viewpoint, but also has considerable practical application value for tackling the socioeconomic effects of typhoons and associated air quality.

## 2 Data and methods

### 2.1 Data

The present study takes 36 air quality monitoring stations in 10 cities in the GBA (Guangzhou, Shenzhen, Zhuhai, Foshan, Zhaoqin, Jiangmen, Huizhou, Dongguan, Zhongshan, Hong Kong) as research objects. The input variables of the model are:

1. The latitude and longitude of the monitoring stations, hourly average AQI and concentrations of $PM_{2.5}$, $PM_{10}$, $SO_2$, $NO_2$ and $O_3$.

2. The typhoon center latitude (Tlat), longitude (Tlon), and minimum pressure (Tpres), as well as the typhoon near-center maximum wind speed (Tws), from the CMA tropical cyclone best-track dataset produced by the CMA Tropical Cyclone Data Center.

3. ERA5 reanalysis meteorological data, including the eastward component of the 10-m wind ($U_{10m}$), northward component of the 10-m wind ($V_{10m}$), 2-m dewpoint temperature ($d_{2m}$) and air temperature ($T_{2m}$), planetary boundary layer





height (PBLH), surface pressure (SP), total precipitation (TP), vertical velocity at 850 hPa ($W_{850}$) and 700 hPa ($W_{700}$), and static stability (St) (defined as the potential temperature at 700 hPa minus that at 1000 hPa).

The distribution of all typhoon samples and air quality monitoring stations is shown in Fig. 1. The data preprocessing procedure can be referred to Text S1.

## 2.2 RF model

The RF algorithm, first proposed by Breiman (2001), is a kind of ensemble machine learning algorithm. The process for establishing the model is follows:

Select a random sample with replacement of the training set and train a large number of decision trees. For each tree, calculate the error at the node and split with the minimum error as the criterion until the designated maximum tree depth is reached. The average of the output of all trees is calculated as the model output.

One of the strengths of the RF model is that it can calculate the importance of features based on impurity, which means that it can calculate the feature's importance by the degree of error reduction brought about by it. The higher the importance value, the more influential the feature. Because of these advantages, RF models have been applied to analyze causal relationships between variables and provide a powerful tool for determining the dominant factors among variables (Wang et al. 2019; Yang et al. 2020; Zeng et al. 2020; Li et al. 2021; Venter et al. 2021; Chen et al. 2022).

Figure 2 presents the technology roadmap for establishing the RF model, which is described as follows:

Step 1. Data acquisition and matching. This paper uses the scikit-learn package in Python to construct the RF forecast model with the typhoon location and intensity data (on typhoon days), the location of monitoring stations and meteorological data as input variables, and the AQI and concentrations of $PM_{2.5}$, $PM_{10}$, $SO_2$, $NO_2$ and $O_3$ as the predicted variables.

Step 2. RF model establishment and cross validation. First, the dataset is randomly divided into a 70% training set and 30%
testing set. The hyperparameter tuning and model training process is conducted on the training set. The hyperparameter tuning process refers to determining the best hyperparameters, which means the parameters must be set manually in advance. The testing set is used for evaluating the RF model's ability to predict the unseen data. To avoid the bias caused by the splitting of the training and testing sets, 10-fold cross-validation (CV) is adopted in the training set. That is, the training set is divided into 10 parts, 9 of which are used as the training set of the tuning process, and then the performance of the remaining
one, called the validation set, is tested. This therefore ensures that the optimal parameters of the model that are found are not affected by data partitioning. The hyperparameters adjusted in the present study are described in Text S2. Afterward, the training set and the testing set were applied to the optimal model respectively, and the feature importance of the model output was analyzed to obtain the dominant meteorological factors of each model.

Step 3. Model prediction and verification. Once the optimal model is established, the training set and testing set are
applied to the model separately, and a series of model evaluation metrics are calculated, including the mean absolute error (MAE), root-mean-square error (RMSE), bias, correlation coefficient between the observed and predicted value ($R$), standard deviation of the observation ($SD_O$), standard deviation of the prediction ($SD_P$), and index of agreement (IA). The definitions



of these metrics are given in Table 1. Among these indicators, the smaller the bias, MAE and RMSE, the better the model performs; the closer $R$ and IA are to 1, the more ideal the result; and the closer $SD_O$ and $SD_P$ are, the better. If RMSE is
lower than $SD_O$, IA is high, and $SD_O$ is close to $SD_P$, the prediction is satisfactory (Lu et al. 1997).

## 3 Results

### 3.1 RF model evaluation

### 3.1.1 TY-associated model

The RF model is applied to the AQI and five pollutants to establish six distinct RF models (the hyperparameters of the six
models can be seen in Table 2; 70% of the samples are used as the training set and 30% as the testing set).

The training and testing results for the AQI, $PM_{2.5}$ and $PM_{10}$ are shown in Fig. 3a, b, d, e, g, h. The $R$ between the observed and predicted value of the training set (testing set) are 0.986 (0.843), 0.986 (0.859) and 0.983 (0.837), respectively; the RMSE are 5.43 (14.88), 3.88 $\mu g/m^3$ (10.31 $\mu g/m^3$) and 6.33 $\mu g/m^3$ (17.03 $\mu g/m^3$); and the bias are 0.10 (-0.07), 0.06 $\mu g/m^3$ (0.20 $\mu g/m^3$) and 0.19 $\mu g/m^3$ (0.16 $\mu g/m^3$). As for the MAE and IA, the RF model also performs well. The IA of the
testing is as high as 0.894, 0.906 and 0.895. It can be seen that the red points in the training set are mostly close to the diagonal line, which means that the RF model makes an accurate prediction over the majority of the samples. Although the data points for the testing set are not as dense as those for the training set, the sample with the most frequency is still relatively close to the $y=x$ line, indicating that the RF model has good predictive ability for unseen data. Concerning the feature importance (Fig. 3c, f, i), the dominant factor of the AQI is $d_{2m}$ which represents the atmospheric humidity, followed
by the static stability. The first two factors have approximate importance values, reflecting that the meteorological determinants of the AQI in the GBA during typhoon events are humidity and static stability. Among all the typhoon information data, the importance of Tlon and Tlat is intermediate among all the variables, while the importance of Tpres and Tws is the lowest. It can be concluded that the typhoon center location rather than the typhoon intensity, is the key to modifying the synoptic situation in the GBA, thereby changing the AQI value. Similarly, Figs. S1–S3 show the $R$ (RMSE)
values of the testing set for $SO_2$, $NO_2$ and $O_3$ are 0.510 (8.13 $\mu g/m^3$), 0.799 (13.64 $\mu g/m^{3)}$ and 0.894 (22.43 $\mu g/m^3$), respectively. The IA of all pollutants except $SO_2$ exceeds 0.85, reflecting that the RF model has strong predictive ability for these pollutants.

The pollutants can be classified into two categories based on the feature importance output of the RF model with respect to the major meteorological controlling factors. The first category is the $d_{2m}$-driven type, which includes $PM_{2.5}$ and $PM_{10}$,
whose dominant meteorological driving factor is $d_{2m}$, followed by the PBLH, which is consistent with the AQI. The reason for this could be that $d_{2m}$ not only reflects the humidity, but also the precipitation and temperature to some extent. When there is rainfall related to a typhoon in the GBA, wet deposition will reduce the concentration of PM. The reason why total precipitation is less important than $d_{2m}$ could be that the latter always has a value and is more variable. The similarity





between the AQI, $PM_{2.5}$ and $PM_{10}$ results reveals that the major pollutant in the ambient air of the GBA during typhoon
events is PM, since the AQI value is the highest of the Individual Air Quality Index. The other category is the PBLH-driven
type, which includes $SO_2$, $NO_2$ and $O_3$. Obviously, the major meteorological influence in this case during typhoon events is
the PBLH. Nevertheless, the situation for $SO_2$ is unlike that of the other two. The most important variables affecting the $SO_2$
concentration after the PBLH are $U_{10m}$ and $V_{10m}$. Indeed, this is the highest $U_{10m}$ and $V_{10m}$ importance among the six models,
indicating that the $SO_2$ in the GBA may mainly derive from transboundary transportation. The variable importance for $NO_2$
and $O_3$ exhibits very similar characteristics because they are both pollutants closely related to photochemical reactions.
Under certain conditions, the free radical reaction of $NO_2$ can generate $O_3$ (Lam et al. 2005, 2018; Zhang et al. 2013; Deng et
al. 2019). It is also worth noting for these two pollutants that the PBLH, which has the highest rank of importance among all
variables, is more than twice as important as the second-highest variable, and this is distinct from the other four models.

Overall, the model has outstanding predictive ability for the AQI and five air pollutants. The present study also highlights
that the typhoon location variables of Tlat and Tlon are more important than the typhoon intensity variables of Tpres and
Tws, showing that the major driving factor in modifying the synoptic situation in the GBA, and thereby changing the AQI
value, is typhoon location. The role of typhoon intensity requires further study. The dominant meteorological drivers of
typhoon-associated air quality are also revealed by the RF model: for the AQI and concentrations of $PM_{2.5}$ and $PM_{10}$ it is $d_{2m}$,
while for $SO_2$, $NO_2$ and $O_3$ it is the PBLH.

**3.1.2 NTY-associated model**

Then, use the meteorological data, station location, and air quality data of the NTY days during the typhoon season (June-
September) to build RF models (the hyperparameters of the six models can be seen in Table 2). Similarly, the model
prediction accuracy and output feature importance are calculated and compared with the results of TY days. The training and
testing results for the AQI, $PM_{2.5}$ and $PM_{10}$ are shown in Fig. 4a, b, d, e, g, h. The $R$ between the observed and predicted
value of the training set (testing set) are 0.979 (0.745), 0.978 (0.744) and 0.978 (0.708), respectively; the RMSE are 5.52
(15.11), 3.60 μg/m³ (9.68 μg/m³) and 5.65 μg/m³ (15.45 μg/m³); and the bias are 0.19 (0.57), 0.12 μg/m³ (0.27 μg/m³) and
0.18 μg/m³ (0.48 μg/m³). Compared with the prediction results of the TY days, the prediction accuracy is significantly
reduced, and the $R$ are all reduced to below 0.8. Figures S4–S6 show the $R$ (RMSE) values of the testing set for $SO_2$, $NO_2$
and $O_3$ are 0.452 (7.00 μg/m³), 0.744 (11.63 μg/m³), 0.867 (24.18 μg/m³), respectively. The prediction accuracy of the model
is significantly poorer compared with the TY days model, and it can be seen that the maximum pollutant concentration on
NTY days is significantly larger than that on TY days, indicating that the period of air quality deterioration in the GBA
coincides with the period of typhoon activity.

The feature importance of model predictions on NTY days is significantly different from that on TY days. For AQI and
$PM_{2.5}$, the meteorological driver is longitudinal wind $V_{10m}$, while for $PM_{10}$ is the latitude of the monitoring station *lat*.
Considering that the southern part of the GBA is close to the sea, and the farther north is, the farther it is from the sea, so
$V_{10m}$ can represent the strength of the sea-land breeze, and *lat* can be seen as the distance from the sea. By contrast, their





meteorological determinants are all $d_{2m}$ on TY days, and this change indicates that the typhoon deters the pollutants from being blown away and replaced by clean air from the ocean, which is the major sink of pollutants on NTY days. Therefore, haze occurs. As for the pollutants classified as the PBLH-driven type, $SO_2$, $NO_2$ and $O_3$. Their meteorological drivers on NTY days are $V_{10m}$, St and PBLH, respectively.

In general, the prediction results indicate that the RF model can accurately and effectively capture the mechanism of the impact of typhoons on air quality. Additionally, differences in meteorological determinants between TY and NTY days also have important implications in air quality in the GBA: for PM, the prevailing sea breeze is the major scavenging mechanism on NTY days, and is deterred by the various wind patterns of the typhoon periphery on TY days. While for $SO_2$, $NO_2$ and $O_3$, on TY days, their concentrations are strongly affected by the PBLH, and the effects of local emission and accumulations are more dramatic than transboundary air pollution, causing polluted events. In contrast, on NTY days, transboundary air pollution is more obvious than the local pollutant emission. These findings shed new light on the control of regional air pollution in the GBA. That is, different strategies should be adopted on TY and NTY days. On NTY days, countermeasures should focus more on source emission control, and make full use of the diffusion and cleaning effect of the sea breeze to reduce air pollution. Coordinated emission reduction in the region should be strengthened to reduce the concentration of pollutants in the entire region at the same time; on TY days, more focus should be on increasing the sink of pollutants (which is decreased due to static and stable weather of the typhoon periphery). Countermeasures should be taken to increase the sedimentation and decomposition of pollutants in the area, such as more road watering.

## 3.2 Model-predicted correlation between air quality and typhoon center location

To further investigate the RF model's ability to capture the correlation between typhoon location and air quality in the GBA, each position within the research area (at a spatial interval of 0.5°) is input into the RF model as the position of the typhoon to predict the AQI and concentrations of $PM_{2.5}$, $PM_{10}$, $SO_2$, $NO_2$ and $O_3$ (the typhoon intensity and meteorological variable values are the averages of all typhoons within the specified spatial interval). Figure 3 shows the average of the predictions across all stations. In all six models, the RF model predicts a low level of air pollution in the GBA when the typhoon is located in the southwest sea area of the GBA, close to Hainan Island. This is because of the relatively clean southerly winds from the sea brought by the cyclonic circulation, large wind speed and precipitation when typhoons are located here. All these meteorological conditions are highly favorable for the deposition and removal of pollutants, and the result is consistent with the findings of previous studies (Yang et al. 2012; Chow et al. 2018; Luo et al. 2018; Yang et al. 2019). By contrast, the air quality in the GBA deteriorates when a typhoon is located over the waters from the Philippines to Taiwan Island, and in the most northerly area over the waters near 30°N. The maximum average concentration of $PM_{10}$ exceeds 80 μg/m³. It is worth noting that the spatial distribution characteristics of the AQI, $PM_{2.5}$ and $PM_{10}$ are very similar because the primary pollutant in the GBA during typhoon weather is likely to be PM, as mentioned above. The distribution of typhoons during $SO_2$ pollution weather is mainly over the sea area to the east of Taiwan Island (16°–27°N), with the maximum $SO_2$ concentration predicted by the model reaching 20 μg/m³. However, the prediction results for $NO_2$ and $O_3$ are scattered,



which may be because their associated photochemical reactions are greatly affected by solar radiation, so the concentrations of these two pollutants possess diurnal variation, which will cause uncertainty in the predictions of the RF model. Nevertheless, the model still accurately captures the overall spatial distribution characteristics; that is, when a typhoon is located over the waters on the southwest side of the GBA, near Hainan Island, the pollutant concentrations are low, but when a typhoon is over the waters near Taiwan Island (117°–125°E), they are high.

## 3.3 Case verification

This paper selects typhoon Danas (2019) as a case to analyze the model's ability to predict typhoon-associated air quality. The active time of Danas was 14–21 July 2019, with a minimum central pressure of 980 hPa. It did not make landfall in China, and its path travelled northwards along the eastern coast of Taiwan Island. During this typhoon event, a significant pollution episode occurred in the GBA (Fig. 6). The synoptic chart shows northerly winds from inland prevailed in the GBA

during the event (July 17–19), which caused pollutants to be transported from inland to the GBA. Meanwhile, the GBA was under high pressure, which was also unfavorable for the diffusion of pollutants (Fig. S7). Figure 7 presents the observed and predicted AQI value and concentrations of $PM_{2.5}$, $PM_{10}$, $SO_2$, $NO_2$ and $O_3$. As Fig. 6a depicts, the track of Danas was L-shaped, which is coincides quite well with the typhoon locations that cause pollution as predicted by the RF model. Around July 16, the typhoon turned north over the sea near the Philippines and then moved along 123°E longitude, gradually

increasing in intensity. The observed data also show a pollution event in the GBA during this period.

First, we examine the spatial distribution of the AQI (Fig. 7a-b). The AQI of the GBA is higher in the northern area than in the southern area during the pollution event. This may be because the southern part is closer to the sea and is affected by a stronger sea breeze, and the RF model successfully predicts this distribution with high accuracy. The distributions of $PM_{2.5}$ and $PM_{10}$ are similar, but the model slightly overestimates their concentrations. The spatial distributions of the $SO_2$, $NO_2$ and

$O_3$ concentrations are relatively scattered and, except for the underestimated concentration of $SO_2$, the predicted results are quite accurate.

Regarding the numerical accuracy of the prediction, Table 3 lists the model evaluation metrics calculated by the average model output. In terms of MAE and RMSE, the largest values are for the predicted $O_3$, which are 15.047 and 18.319 $\mu g/m^3$, respectively. Meanwhile, the smallest MAE (RMSE) is found for $PM_{2.5}$ ($SO_2$), which is 4.117 (4.876) $\mu g/m^3$. The $R$ values

between the observations and predictions of the AQI and five pollutants all exceed 0.7, with that of the AQI and $O_3$ even exceeding 0.85. The bias values of the predicted AQI and five pollutants are all less than 0, indicating that the RF model tends to underestimate in this case. The RMSEs of the result of the AQI, $PM_{2.5}$, $PM_{10}$, $NO_2$ and $O_3$ are lower than the $SD_O$ values, and the $SD_O$ and $SD_P$ values of all the pollutants are quite close. Furthermore, the IA is high. Among all the models, the IA of the AQI, $PM_{2.5}$ and $O_3$ exceeds 0.9, indicating that these three air quality parameters perform the best in this case.

The evaluation metrics of the results in 10 cities are listed in Tables S1–S10, revealing that 39 (66%) of all air quality parameter predictions in these cities have an RMSE less than the $SD_O$, and 31 (53%) have an IA exceeding 0.8. Generally, the best-performing pollutants are $PM_{2.5}$ and $O_3$, as judged by the metrics, while the performance with respect to $SO_2$ needs





improvement. The MAE and RMSE values obtained by city are both larger than the values obtained by the average over the entire GBA, because the averaging process eliminates some random errors.

In summary, the evaluation metric results are extremely encouraging, and indicate a satisfactory prediction by the RF model of the Danas-associated air quality in the GBA. Moreover, the RF model obtains temporal information from the diurnal variation of the input features such as typhoon intensity to accurately predict the diurnal fluctuations of $NO_2$ and $O_3$, which reflects the model's ability to capture the nonlinear relationship and its potential for tackling complex prediction problems.

## 4 Conclusions and discussions

Typhoons are highly active weather systems in summer that have substantial effects on the synoptic situation in the entire southern part of China, including the Guangdong–Hong Kong–Macao Greater Bay Area. In addition to causing violent winds, rainfall and storm surges in the area close to their location, typhoons also affect the background circulation situation in areas more distant from their immediate vicinity. For instance, the typhoon periphery downdraft brings about light winds,
stagnant weather, high temperatures, and a low planetary boundary layer, and consequently have a detrimental impact on the generation, transportation and diffusion of air pollutants, causing hazy weather. The Guangdong–Hong Kong–Macao Greater Bay Area, located at the southernmost tip of the Chinese mainland, is often affected by typhoons. Therefore, air pollution events associated with typhoons in the GBA are prevalent in summer.

The present study employs the RF model to predict the typhoon-associated air quality quantitatively. The $R$ (RMSE)
values of the testing set for the AQI, $PM_{2.5}$, $PM_{10}$, $SO_2$, $NO_2$ and $O_3$ are 0.843 (14.88), 0.859 (10.31 µg/m³), 0.837 (17.03 µg/m³), 0.510 (8.13 µg/m³), 0.799 (13.64 µg/m³) and 0.894 (22.43 µg/m³), respectively. The results are satisfactory overall. Then, the model is verified using the case of typhoon Danas (2019). The results are averaged over the GBA, and the $R$ (RMSE) values of the AQI, $PM_{2.5}$, $PM_{10}$, $SO_2$, $NO_2$ and $O_3$ are 0.862 (7.458), 0.841 (5.136 µg/m³), 0.793 (8.135 µg/m³), 0.727 (4.876 µg/m³), 0.827 (5.633 µg/m³) and 0.952 (18.319 µg/m³), respectively. The prediction is accurate for both the air
quality of one city and the average air quality in the GBA. In contrast, using meteorological data to predict the air quality of NTY days, the accuracy is significantly lower than the results of TY days, indicating that the impact mechanism of typhoons on air pollution is accurately captured by the model, and it is important for the improvement of model prediction accuracy.

Another important finding of the present study is that the difference in feature importance output by the RF model on TY days and NTY days. On TY days, the meteorological driver of AQI, $PM_{2.5}$ and $PM_{10}$ is the $d_{2m}$ that represents the air
humidity, while $SO_2$, $NO_2$ and $O_3$ are dominated by the height of the boundary layer. Differently, on NTY days, their dominant meteorological factors were changed, and the importance of variables representing regional transportation and sea breeze diffusion was significantly higher than that in TY days. These findings suggest that the prevailing sea breeze is the major scavenging mechanism of pollutants on NTY days, and is deterred by the various wind patterns of the typhoon periphery on TY days. This implies that different control strategies should be adopted on TY days and NTY days. On TY



days, countermeasures should be taken to increase the sink of pollutants in order to compensate for the effect of the weakened sea breeze and the hindered diffusion of pollutants caused by the static and stable weather of the typhoon periphery.

Besides, the present study also highlights the following:

1. The feature importance output by the RF model indicates that the typhoon location is more important than the intensity,

suggesting that the most significant factor in modifying the synoptic condition, and thereby changing the air quality, is the location of the typhoon center.

2. By sampling at a spatial interval of 0.5° and inputting the data into the RF model as the typhoon center location, the prediction result is consistent with previous studies; that is, the air quality in the GBA deteriorates when the typhoon passes over the waters near Taiwan Island.

3. The concentrations of $NO_2$ and $O_3$ possess diurnal variation as a result of their photochemical reactions in the atmosphere, and the RF model predicts this diurnal cycle with high accuracy because of the diurnal variation of the input variables such as air temperature, PBLH, typhoon intensity and wind speed.

Overall, the RF model achieves good results in predicting typhoon-associated air quality. Compared with approaches adopted in previous research, such as numerical simulation and statistical modelling, the RF model has the advantages of

high accuracy and convenient application, and produces a precise quantitative prediction of typhoon-associated air quality in the GBA. At the same time, the importance of features revealed by the model also shed new light on regional pollution control on typhoon days. Of course, the impact of typhoons on air quality is not limited to the GBA, but the model structure provided in the present study can be applied conveniently to various areas, which gives it significant application value for air pollution prevention and control. It is worth mentioning that not all typhoons affect the air quality in their area of impact,

because of the substantial variability of typhoon tracks. The $R$ and RMSE values in the case study are better than those of the whole dataset, reflecting that some typhoons in the dataset do not directly affect the air quality in the GBA. Meanwhile, as mentioned earlier, the air quality is also affected by factors such as source emissions. The RF model's prediction of the air quality in the GBA under these scenarios merits further study.

*Code availability.* The model in this paper is based on the scikit-learn package in Python, and the implementation and

analysis code are available upon request to the corresponding author (yyj1985@nuist.edu.cn).

*Data availability.* The datasets that are analyzed and used to support the findings of this study are available in the public domains: The air quality observation data are deposited at linkage: https://doi.org/10.5281/zenodo.6945344 (Chen 2022), which are provided by the China National Environmental Monitoring Center and the Environmental Protection Interactive Centre of the Environmental Protection Department, Hong Kong Special Administrative Region government. The CMA

tropical cyclone best-track dataset can be obtained from the CMA Tropical Cyclone Data Center



(https://tcdata.typhoon.org.cn/en/zjljsjj_zlhq.html). The ERA5 reanalysis data set is available at the European Centre for Medium-Range Weather Forecasts (https://doi.org/10.24381/cds.adbb2d47 and https://doi.org/10.24381/cds.bd0915c6).

*Author contributions.* YY was responsible for conceptualization, supervision and funding acquisition. YC were responsible for data curation, programing and original draft preparation. MG made critical comments on this study. YC and YY developed the methodology and carried out formal analysis. YC, MG and YY were responsible for results discussion, text review and editing.

*Competing interests.* The authors declare that they have no conflict of interests.

*Financial support.* This research has been supported by the National Natural Science Foundation of China (grant no. 42175098).

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



**Table 1. The definition of evaluation metrics of the model.**

| Metric | Definition |
|--------|-----------|
| MAE | $\dfrac{1}{N}\sum_{i=1}^{N} |\phi_i|$ |
| RMSE | $\left[\dfrac{1}{N}\sum_{i=1}^{N} (\phi_i)^2\right]^{1/2}$ |
| Bias | $\dfrac{1}{N}\sum_{i=1}^{N} \phi_i$ |
| $R$ | $\dfrac{\sum_{i}^{N} (O_i - \bar{O})(p_i - \bar{p})}{\sqrt{\sum_{i=1}^{N} (O_i - \bar{O})^2}\sqrt{\sum_{i=1}^{N} (p_i - \bar{p})^2}}$ |
| SD$_O$ | $\dfrac{1}{N-1}\left[\sum_{i=1}^{N} (O_i - \bar{O})^2\right]^{1/2}$ |
| SD$_P$ | $\dfrac{1}{N-1}\left[\sum_{i=1}^{N} (p_i - \bar{p})^2\right]^{1/2}$ |
| IA | $1 - \dfrac{\sum_{i=1}^{N} (\phi_i)^2}{\sum_{i}^{N} (|p_i - \bar{O}| + |O_i - \bar{O}|)^2}$ |

Notation: $p_i$ is the predicted value; $O_i$ is the observed value; $N$ is sample size; $\bar{O}$ is the mean of observed value; $\bar{p}$ is the mean of predicted value; $\phi_i$ is the difference between the predicted and observed values.






**Table 2. The best hyper-parameters of the model.**

| Model | n-estimator | | max-depth | |
|---|---|---|---|---|
| | TY days | NTY days | TY days | NTY days |
| AQI | 710 | 750 | 82 | 87 |
| $PM_{2.5}$ | 170 | 630 | 70 | 88 |
| $PM_{10}$ | 420 | 690 | 150 | 41 |
| $SO_2$ | 250 | 385 | 72 | 61 |
| $NO_2$ | 580 | 685 | 100 | 71 |
| $O_3$ | 660 | 495 | 80 | 101 |




**Table 3. Evaluation metrics of the model prediction of the case Danas in the GBA.**

|  | AQI | PM$_{2.5}$ | PM$_{10}$ | SO$_2$ | NO$_2$ | O$_3$ |
|---|---|---|---|---|---|---|
| MAE | 5.470 | 4.117µg/m$^3$ | 6.222µg/m$^3$ | 4.529µg/m$^3$ | 5.037µg/m$^3$ | 15.047µg/m$^3$ |
| RMSE | 7.458 | 5.136µg/m$^3$ | 8.135µg/m$^3$ | 4.876µg/m$^3$ | 5.633µg/m$^3$ | 18.319µg/m$^3$ |
| Bias | -2.265 | -1.453µg/m$^3$ | -1.509µg/m$^3$ | -4.529µg/m$^3$ | -0.769µg/m$^3$ | -3.870µg/m$^3$ |
| *R* | *0.862 | *0.841 | *0.793 | *0.727 | *0.827 | *0.952 |
| SD$_O$ | 10.705 | 7.900µg/m$^3$ | 11.139µg/m$^3$ | 1.451µg/m$^3$ | 9.332µg/m$^3$ | 47.514µg/m$^3$ |
| SD$_P$ | 10.650 | 7.323µg/m$^3$ | 10.679µg/m$^3$ | 1.921µg/m$^3$ | 6.794µg/m$^3$ | 41.153µg/m$^3$ |
| IA | 0.917 | 0.906 | 0.884 | 0.452 | 0.881 | 0.966 |

Note: the correlation coefficient marked with "*" is significant with a significance level of 0.05.



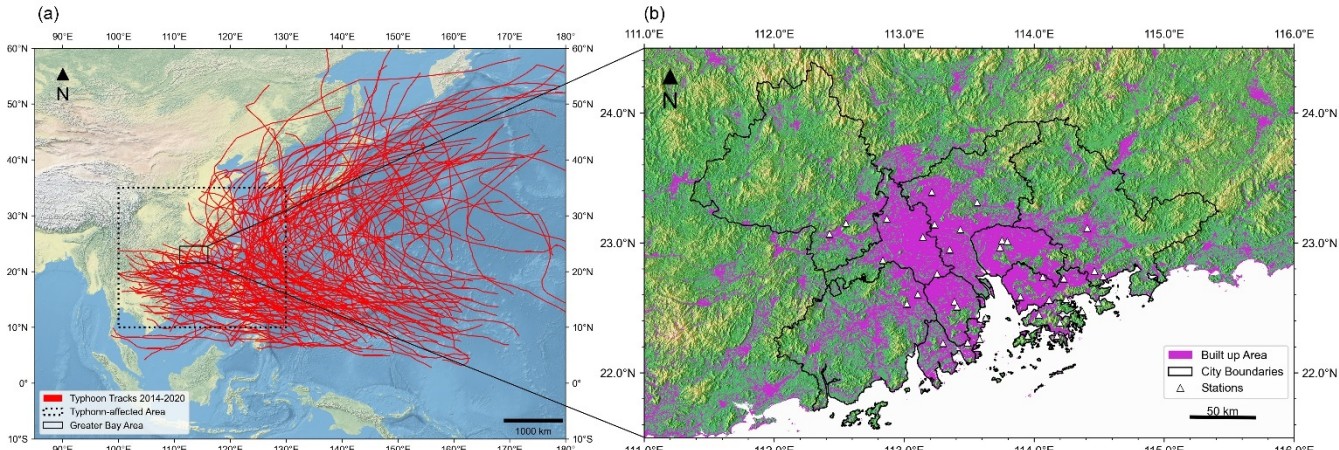

**Figure 1. Overview of the data used in this study: (a) tracks of the studied typhoons (only those typhoons within the dotted box area are introduced into the model); (b) locations of the 36 observation stations.**



**Step 1: Data acquisition and matching**

Input variables

TY days:

Typhoon center longitude
Typhoon center latitude
Typhoon center minimum pressure
Typhoon near-center maximum wind speed

NTY days:

ERA5 reanalysis metorological data:
$U_{10 m}$, $V_{10 m}$, $d_{2 m}$, $T_{2 m}$, PBLH, SP, TP, $W_{850}$, $W_{700}$, St

+

Predicted variables

AQI, $PM_{2.5}$, $PM_{10}$, $SO_2$, $NO_2$, $O_3$

Longitude and latitude of
observation stations

| Station ID | Lon | Lat |
|---|---|---|
| 1345A | 113.23 | 23.14 |
| 1349A | 113.43 | 23.10 |
| 1350A | 113.35 | 22.95 |
| ... | ... | ... |
| HK5 | 114.18 | 22.28 |

Match in space and time

**Step 2: Random forest model establishment and cross validation**

Splitting training and testing dataset

70% Training set

Applying 10-fold CV to determine
the best parameters of RF model

··· ···

Establishing RF model

Decision tree × n

30% Testing set

Applying testing set to the best model

**Determining the main modulating factors via RF model features importance**

**Step 3: Model prediction and verification**

MAE
RMSE
R
etc.

Calculating evaluation metrics

for both training and testing set

Model verification with
real typhoon case

**Figure 2. Flow chart of the study framework.**








**Figure 3. The result of TY days' AQI, PM$_{2.5}$ and PM$_{10}$ predicted by the RF model. (a) training set of AQI; (b) testing set of AQI; (c) feature importance of AQI; (d-i) training set of (d) PM$_{2.5}$ and (g) PM$_{10}$; testing set of (e) PM$_{2.5}$ and (h) PM$_{10}$; feature importance of (f) PM$_{2.5}$ and (i) PM$_{10}$.**




**Figure 4. The result of NTY days' AQI, PM$_{2.5}$ and PM$_{10}$ predicted by the RF model. (a) training set of AQI; (b) testing set of AQI; (c) feature importance of AQI; (d-i) training set of (d) PM$_{2.5}$ and (g) PM$_{10}$; testing set of (e) PM$_{2.5}$ and (h) PM$_{10}$; feature importance of (f) PM$_{2.5}$ and (i) PM$_{10}$.**





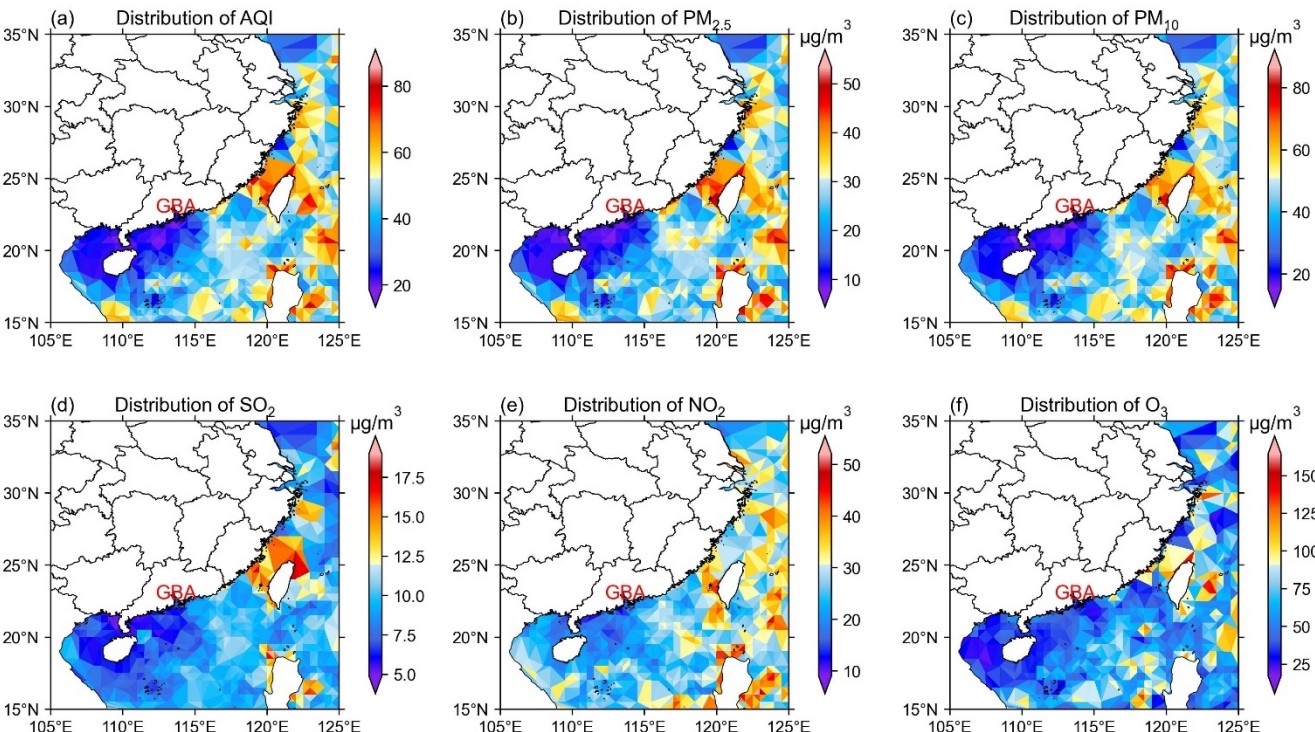

**Figure 5.** The correlation of air quality over the Greater Bay Area and typhoon center location predicted by the model. (a) the correlation of AQI and typhoon center location predicted by the model. The shading indicates the average value of air quality when the typhoon locates in the corresponding location; (b) PM2.5; (c) PM10; (d) SO2; (e) NO2; (f) O3.



**Figure 6. Track of typhoon Danas (2019) and the observed and model-predicted air quality (the value of a city is the mean value of all its stations): (a) track and minimum pressure of typhoon Danas from 2000 BJT 15 July 2019 to 1400 BJT 20 July 2019; (b) the observed AQI value; (c) the model-predicted AQI value; (d) the observed PM$_{2.5}$ concentration; (e) the model-predicted PM$_{2.5}$ concentration; (f) the observed PM$_{10}$ concentration; (g) the model-predicted PM$_{10}$ concentration; (h) the observed SO$_2$ concentration; (i) the model-predicted SO$_2$ concentration; (j) the observed NO$_2$ concentration; (k) the model-predicted NO$_2$ concentration; (l) the observed O$_3$ concentration; (m) the model-predicted O$_3$ concentration.**

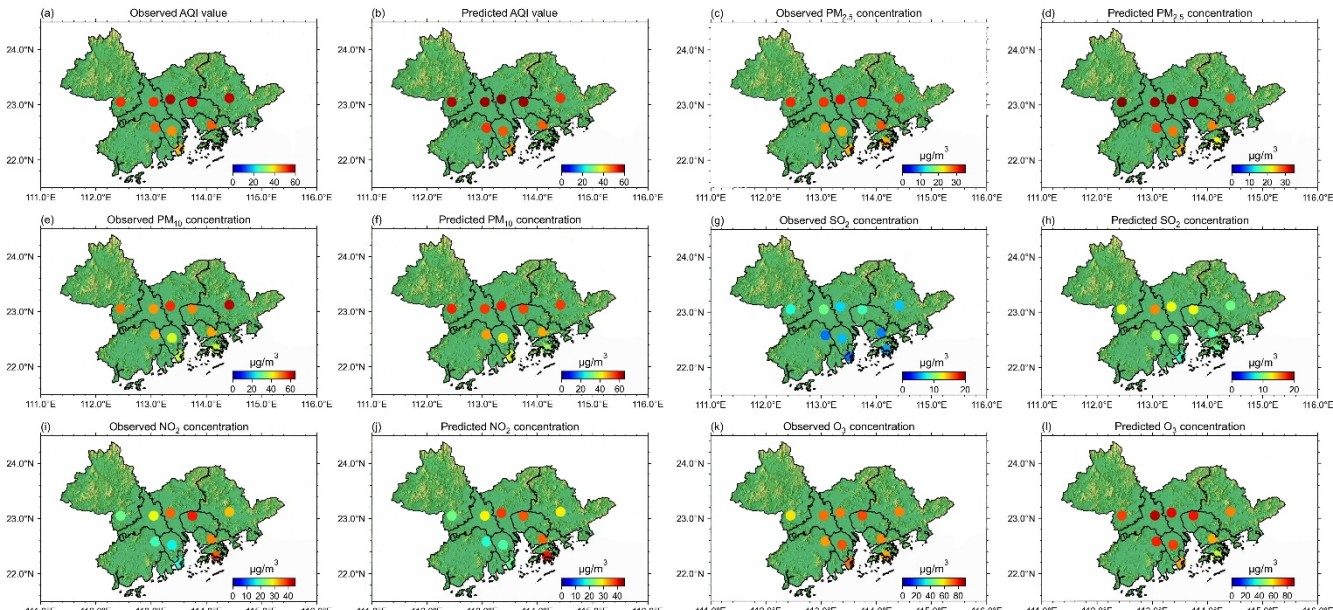

**Figure 7. The spatial distribution of AQI and five pollutants during typhoon Danas. (a) observed AQI value; (b) predicted AQI value; (c-d) PM$_{2.5}$; (e-f) PM$_{10}$; (g-h) SO$_2$; (i-j) NO$_2$; (k-l) O$_3$.**