# Peer review of "Typhoon-associated air quality over the Guangdong-Hong Kong-Macao Greater Bay Area, China: machine learning-based prediction and assessment"

_Atmospheric Measurement Techniques, 2022_

## Author Response (AR1)

**Response to reviewer comments**

We are sincerely grateful to the editor and reviewers for their valuable time spent on reviewing our manuscript. The comments are very helpful and valuable, and we have addressed the issues raised by the reviewers in the revised manuscript. Please find our point-by-point response (in blue font) to the comments (in black font) raised by the reviewers.

Referee #1:

COMMENTS TO THE AUTHOR(S)

This paper used the random forest models to predict typhoon-associated air quality in the Guangdong-Hong Kong-Macao Greater Bay Area. This work is interesting and meaningful to the readers of AMT. Generally this paper is well organized. The methods and conclusions are reliable. But I still have some suggestions below.

Response: Thank you so much for your efforts for reviewing our manuscript. The comments are very constructive, and we have now further revised our manuscript in light of these comments.

Detailed comments:

1. This paper used data from 36 air quality monitoring stations in 10 cities of the GBA that is heavily polluted. Why did you not consider the observation data from the rural regions of Guangdong Province?

Response: Many thanks for your kind suggestion. Relative to rural areas, both the population and stations are denser over urban area. Our objective is to explore the performance of the machine learning model over urban areas in the Guangdong–Hong Kong–Macao Greater Bay Area. We hope this study can help improve the prediction and assessment of typhoon-associated air quality over cities in the GBA.

Are the 36 air quality monitoring data are used for both model training and RF model evaluation? I think it may be more reasonable if you could choose parts of the 36 sites for model training and the others for RF model evaluation.

Response: Thank you for your comments. Yes, the data for 36 monitoring stations are used for both RF model training and testing. We divide the whole dataset into the testing set and training set to ensure that model will not see the training data in the testing stage, and 10-fold cross-validation is applied to minimize the data proportion bias. Besides, we employed an independent sample during typhoon Danas (outside of the training and testing dataset) to test the RF model's generalization ability. Per your suggestion, we added more samples from three new stations to evaluate RF model (lines 87-88). Figure 4 shows the result of three testing stations in Guangzhou, Shenzhen and Hong Kong on both TY and NTY days. Lines 176-179 in section 3.1.1 show the results for these three stations, the $R$ (RMSE) values for AQI, $PM_{2.5}$, $PM_{10}$, $SO_2$, $NO_2$ and $O_3$ on TY days are 0.868(11.70), 0.900 (7.16 μg/m$^3$), 0.841 (13.45 μg/m$^3$), 0.496 (5.38 μg/m$^3$), 0.538 (27.94 μg/m$^{3)}$ and 0.878 (22.45 μg/m$^3$), respectively. The result indicates that the RF model successfully captures the correlation between typhoon's location and monitoring stations' location. Though the input stations changed, the model still produces accurate prediction based on the relative position of the station and the

typhoon.

[Figure]

The results of $SO_2$, $NO_2$ and $O_3$ are shown in Fig. S4, which is consistent with the results of AQI, $PM_{2.5}$, $PM_{10}$.

2. In section 3.1, did you compare the model performance of the RF model with the other traditional air quality models, e.g., CMAQ, WRF/Chem.

Response: Thanks for your suggestion. Compared with traditional numerical approach, ML methods have the advantage of lower computational cost, which give ML models huge application potential. Therefore, we aim to explore the ability and application of the machine learning model in the present study. The numerical models such as CMAQ, and WRF/Chem will be compared with RF model in our future work.

Figure 5: Why the data are only available over the seas?

Response: Thank you for comment. We followed your suggestion and edited the figure. The data are available in the land now, and the results are illustrated by scattering points instead of shading as following. Because we added a new figure, it is Fig. 6 now.

[Figure]

Figure 6: The same data at the air quality monitoring sites are first used for model training and then for model evaluation?

Response: Thanks for your suggestion. They are different and independent. We have added this point at lines 256-258 in section 3.3 as follows:

This paper takes typhoon Danas (2019) as an independent case to analyze the model's ability to predict typhoon-associated air quality over the GBA. For better evaluation of RF model, typhoon Danas's data have been removed from the dataset in training and testing steps.

Referee #3:

COMMENTS TO THE AUTHOR(S)

This study employs the random forest models to predict typhoon-associated air quality quantitatively in the Guangdong-Hong Kong-Macao Greater Bay Area. The prediction models are established for typhoon and non-typhoon days. Thus, the results suggest that different air pollution control strategies for typhoon days and non-typhoon days should be adopted. The work is innovative well written and interesting to the readers of AMT.

I have two questions below.

1. The present study takes 36 air quality monitoring stations in 10 cities in the GBA (Guangzhou, Shenzhen, Zhuhai, Foshan, Zhaoqin, Jiangmen, Huizhou, Dongguan, Zhongshan, Hong Kong) as research objects. Why did you not consider data from the rural regions?

Response: Many thanks for your efforts for reviewing our manuscript and your kind suggestion. Both the population and stations are denser over urban area relative to rural areas. Our objective is to explore the performance, indication and possible application of machine learning-based algorism in typhoon associated air quality in the Guangdong–Hong Kong–Macao Greater Bay Area. We hope this study can help improve the prediction and assessment of typhoon-associated air quality over cities in the GBA.

2. The study used ERA5 reanalysis from meteorological data. Couldn't these data be integrated with those coming from other high-resolution instruments? For example, lidar or meteorological radiosondes?

Response: Thank you for the question. Indeed, the inputs of observations from other high-resolution instruments (e.g., lidar and radiosondes) can help improve the accuracy of RF model. Nevertheless, we cannot obtain lidar or meteorological radiosondes in the GBA in the present stage. More importantly, by using the ERA5 reanalysis, we can introduce the ECMWF's forecast data into the model, along with the predicted typhoon tracks and intensity (such as that released by the CMA), into our model to make the prediction. We stated this point at lines 99-101 in section 2.1 as follows:

Using the model constructed with the data above, the future air quality under the effect of the typhoon can be predicted. To be specific, the forecasted air quality can be acquired by replacing the ERA5 reanalysis meteorological data with the ECMWF's forecast field and introducing the predicted typhoon location and intensity (for example, from the CMA).